# New Meloxicam Derivatives—Synthesis and Interaction with Phospholipid Bilayers Measured by Differential Scanning Calorimetry and Fluorescence Spectroscopy

**DOI:** 10.3390/membranes13040416

**Published:** 2023-04-06

**Authors:** Jadwiga Maniewska, Justyna Gąsiorowska, Żaneta Czyżnikowska, Krystyna Michalak, Berenika M. Szczęśniak-Sięga

**Affiliations:** 1Department of Medicinal Chemistry, Faculty of Pharmacy, Wroclaw Medical University, Borowska 211, 50-556 Wrocław, Poland; 2Department of Biophysics and Neuroscience, Wroclaw Medical University, T. Chałubińskiego 3a, 50-368 Wrocław, Poland; 3Department of Basic Chemical Sciences, Faculty of Pharmacy, Wroclaw Medical University, Borowska 211a, 50-556 Wrocław, Poland

**Keywords:** model membranes, DPPC, DSC, fluorescence spectroscopy, laurdan, prodan, 1,2-benzothiazine derivatives, synthesis, molecular modeling, oxicams, meloxicam analogues, drug–membrane interaction

## Abstract

The purpose of the present paper was to assess the ability of five newly designed and synthesized meloxicam analogues to interact with phospholipid bilayers. Calorimetric and fluorescence spectroscopic measurements revealed that, depending on the details of the chemical structure, the studied compounds penetrated bilayers and affected mainly their polar/apolar regions, closer to the surface of the model membrane. The influence of meloxicam analogues on the thermotropic properties of DPPC bilayers was clearly visible because these compounds reduced the temperature and cooperativity of the main phospholipid phase transition. Additionally, the studied compounds quenched the fluorescence of prodan to a higher extent than laurdan, what pointed to a more pronounced interaction with membrane segments close to its surface. We presume that a more pronounced intercalation of the studied compounds into the phospholipid bilayer may be related to the presence of the molecule of a two-carbon aliphatic linker with a carbonyl group and fluorine substituent/trifluoromethyl group (compounds **PR25** and **PR49**) or the three-carbon linker together with the trifluoromethyl group (**PR50**). Moreover, computational investigations of the ADMET properties have shown that the new meloxicam analogues are characterized by beneficial expected physicochemical parameters, so we may presume that they will have a good bioavailability after an oral administration.

## 1. Introduction

The interaction of the drugs with the phospholipid bilayer is crucial in many biochemical processes [1]. Model membranes are often used to assess such interactions. These may be the systems that imitate the organization of lipids in the natural biological membrane [2]. The use of model membranes provides a lot of important information about the drug localization in the bilayer, as well as about its influence on the properties of the membrane. A drug may act on the surface of the membrane or have intracellular targets; therefore, in order to elicit a response, they must penetrate the lipid bilayer. In order to be absorbed, distributed, metabolized, and eliminated, the drug has to pass through the biological membrane, which ultimately affects the effectiveness of the therapeutic agents used [3,4].

The use of membrane models, which may include, e.g., micelles, liposomes, lipid monolayers, and lipid bilayers on a carrier, provides information on the effect of studied drugs on the membrane. The use of micelles showed that there is a relationship between the interaction of the drug with the membrane and the length of the hydrophobic lipid chain and the surface charge of biological membranes [3].

In this paper, new meloxicam analogues and their interaction with model phospholipid membranes are discussed. Meloxicam belongs to a class of medicines, known as non-steroidal anti-inflammatory drugs (NSAIDs) [5]. Their biological target is cyclooxygenase (COX), a membrane protein associated with the phospholipid bilayer surrounding the endoplasmic reticulum, as well as the cell nucleus [6,7]. Drugs in this group are mainly taken orally; therefore, drug–membrane interaction is a preliminary stage in the body as drugs must cross biological membranes in order to be absorbed, then they are distributed, and, finally, they reach their biological target [8].

The main lipid-based model systems, which are biomimetic systems—the models of biological membranes—are liposomes, phospholipid bilayers, Langmuir monolayers at the air–water interface, supported monolayers, and micelles [9,10,11,12,13]. The study of NSAID–membrane interactions using liposomes enables the assessment of the effect of the drug on the phase transition temperature of lipids, as well as on the fluidity and packing of the membrane. Using liposomes, the logP and the location of NSAIDs in the bilayer may also be determined. Lipid monolayers and supported lipid bilayers are used to analyze modifications of the structure of lipids as a result of membrane–drug interactions [3].

Studies on the interaction of NSAIDs with the lipid membrane can provide information on the effects of drugs, both therapeutic and toxic [14]. Models built of phosphatidylcholine are most often used to study the interaction of non-steroidal anti-inflammatory drugs with the membrane because it is the main structural lipid of the membrane. Phosphatidylcholine is abundant in the mucous membrane of the digestive tract, where it has a protective effect on the mucosa [15]. Therefore, it is often chosen to assess the effects of local NSAIDs on the gastrointestinal tract. What is more, phosphatidylcholine is the main component of the lipid layer of the synovial fluid, while NSAIDs are often used in the treatment of pain and inflammation associated with musculoskeletal disorders [3,4].

In the present work, we describe the synthesis and results of the calorimetric and fluorescence spectroscopic studies of five new analogues of meloxicam (**PR23**, **PR24**, **PR25**, **PR49,** and **PR50**; see Figure 1 and Figure 1) on the phase behavior of phospholipid bilayers and fluorescence quenching of two fluorescent probes, laurdan and prodan. In new meloxicam derivatives, the substituents in position 2 and 3 of the 1,2-benzothiazine scaffold have been modified in order to obtain better pharmacological parameters and reduced toxicity. In position 2, an extended phenylpiperazine substituent was introduced in place of the small methyl group in meloxicam (see Figure 1). While in position 3, the thiazole carboxamide moiety was replaced with a benzoyl moiety, because Wiesław Malinka et al.’s research showed that it enhances the analgesic activity of the compound [16]. Similar 1,2-benzothiazine derivatives were previously synthesized, and their biological activity was tested [17]. It occurred that they showed analgesic activity in an animal model, and this was also the reason why we used meloxicam, a well-known non-steroidal anti-inflammatory drug (NSAID) [18] consisting 1,2-benzothiazine scaffold as a reference compound.

The results presented allow us to draw the conclusion that the studied compounds interact with the phospholipid bilayers under consideration. Due to the fact that studied compounds may be considered future drug candidates, the ADMET parameters of new meloxicam analogues were also estimated.

## 2. Materials and Methods

### 2.1. Chemicals

Phospholipid: 1,2-dipalmitoyl-n-glycero-3-phosphatidylcholine (DPPC) was obtained from Merck KGaA (Darmstadt, Germany). It was used as delivered without any purification. The same phospholipid was used in calorimetric and spectroscopic studies. Tris-EDTA buffer solution was purchased from Merck KgaA (Darmstadt, Germany). Meloxicam was purchased from Alfa Aesar (Karlsruhe, Germany).

Fluorescent labels: *N*,*N*-dimethyl-6-dodecanoyl-2-naphthylamine (laurdan) and *N*,*N*-dimethyl-6-propionyl-2-naphthylamine (prodan) were purchased from Molecular Probes (Thermo Fisher Scientific, Eugene, TX, USA). Fluorescent probes were dissolved in dimethyl sulfoxide (DMSO) to obtain 1 mM stock solutions. Since the studied compounds were insoluble in water, their chloroform or DMSO solutions were used for the experiments. All other chemicals used in this study were of an analytical grade.

### 2.2. Experimental

#### 2.2.1. Synthesis

The melting points were recorded using MEL-TEMP capillary melting point apparatus and were uncorrected. ^1^H NMR spectra were measured on a Brucker 300 MHz NMR spectrometer using TMS as an internal reference. The samples were prepared by dissolving in chloroform. Chemical shift (*δ*) values are given in parts per million (ppm). Splitting patterns are designated as follows: s, singlet; brs, broad singlet; d, doublet; t, triplet; q, quartet; m, multiplet. IR spectra (KBr, cm^−1^) were recorded on a Specord-75 IR spectrometer. The elemental C, H, and N analyses were performed by a Carlo Erba NA-1500 analyzer and were within ±0.4% of the values calculated for the corresponding formulas. The progress of the reactions and the purity of the prepared compounds were monitored by thin-layer chromatography (TLC) using Fluka pre-coated aluminum plate silica gel with a fluorescent indicator of 254 nm. Reagents and solvents were purchased from commercial suppliers and were used as received.

The synthesis and experimental data of compounds **3a,b** and **4a,b** were previously reported [17,19], as well as piperazine derivatives **5**, **6**, **7,** and **8** [20,21,22].

***Synthesis pathway and experimental data of new compounds* PR23, PR24, PR25, PR49**, and **PR50** (see Figure 1).

To the stirred mixture of 5 mmol of compound **4a** or **4b** in 20 mL of anhydrous ethanol was added 5 mL of sodium ethanoate (2.3%). Then, 5 mmol of compound **5** (for **PR49**) or compound **6** (for **PR50**) or compound **7** (for **PR24**) or compound **8** (for **PR23** and **PR25**) was added and refluxed with stirring for 8–10 h. When the reaction ended, which was controlled on TLC plates, ethanol was distilled off, the residue was treated with 50 mL of chloroform, and the insoluble materials were filtered off. The filtrate was then evaporated, and the residue was purified by crystallization from ethanol to produce pure products.

**PR23 *2-{2-[4-(o-fluorophenyl)-1-piperazinyl]-2-oxoethyl}-4-hydroxy-3-(4-methoxybenzoyl)**-2H-1,2-benzothiazine 1,1-dioxide*** yellow powder, 45% yield, mp 110–112 °C. IR (KBr, cm^−1^): 1665, 1600 (CO), 1345, 1180 (SO_2_). ^1^H NMR (300 MHz, CDCl_3_) *δ* (ppm): 1.68 (brs, 2H, C**H_2_**CO), 2.82–2.90 (m, 4H of piperazine), 3.29 (brs, 4H of piperazine), 3.90 (s, 3H, OCH_3_), 6.83–8.26 (m, 12H, ArH), 15.69 (s, 1H, OH*_enolic_*). Anal. Calcd for C_28_H_26_FN_3_O_6_S (551.58): C, 60.97; H, 4.75; N, 7.62; found: C, 60.75; H, 4.68; N, 7.42.

**PR24 *2-{3-[4-(o-fluorophenyl)-1-piperazinyl]propyl}-4-hydroxy-3-(4-methylbenzoyl)**-2H-1,2-benzothiazine 1,1-dioxide*** yellow crystals, 48% yield, mp 155–158 °C. IR (KBr, cm^−1^): 1610 (CO), 1345, 1170 (SO_2_). ^1^H NMR (300 MHz, CDCl_3_) *δ* (ppm): 1.26 (brs, 2H, CH_2_C**H_2_**CH_2_), 1.98 (brs, 2H, C**H_2_**CH_2_CH_2_N-*piperazine*), 2.27 (brs, 4H of piperazine), 2.45 (s, 3H, CH_3_), 2.95–3.39 (m, 6H; 2H of CH_2_CH_2_C**H_2_** and 4H of piperazine), 6.87–8.21 (m, 12H, ArH), 15.70 (s, 1H, OH*_enolic_*). Anal. Calcd for C_29_H_30_FN_3_O_4_S (535.63): C, 65.03; H, 5.65; N, 7.84; found: C, 65.35; H, 5.35; N, 7.83.

**PR25 *2-{2-[4-(o-fluorophenyl)-1-piperazinyl]-2-oxoethyl}-4-hydroxy-3-(4-methyl-benzoyl)**-2H-1,2-benzothiazine 1,1-dioxide*** yellow powder, 52% yield, mp 148–149 °C. IR (KBr, cm^−1^): 1660, 1610 (CO), 1340, 1180 (SO_2_). ^1^H NMR (300 MHz, CDCl_3_) *δ* (ppm): 2.45 (s, 3H, CH_3_), 2.89–4.28 (m, 10H; 2H of C**H_2_**CO and 8H of piperazine), 6.86–8.24 (m, 12H, ArH), 15.66 (s, 1H, OH*_enolic_*). Anal. Calcd for C_28_H_26_FN_3_O_5_S (535.59): C, 62.79; H, 4.89; N, 7.85; found: C, 63.05; H, 5.00; N, 7.84.

PR49 *2-{2-[4-(m-trifluoromethylphenyl)-1-piperazinyl]-2-oxoethyl}-4-hydroxy-3-(4-methyl-benzoyl)-2H-1,2-benzothiazine 1,1-dioxide* orange powder, 7% yield, mp 113–116 °C. IR (KBr, cm^−1^): 1675, 1610 (CO), 1350, 1180 (SO_2_). ^1^H NMR (300 MHz, CDCl_3_) *δ* (ppm): 2.43 (s, 3H, CH_3_), 2.99–4.12 (m, 10H; 2H of CH_2_CO and 8H of piperazine), 7.09–8.27 (m, 12H, ArH), 15.65 (s, 1H, OH*_enolic_*). Anal. Calcd for C_29_H_26_F_3_N_3_O_5_S (585.59): C, 59.48; H, 4.48; N, 7.18; found: C, 59.18; H, 4.26; N, 6.90.

**PR50 *2-{3-[4-(m-trifluoromethylphenyl)-1-piperazinyl]propyl}-4-hydroxy-3-(4-methyl-benzoyl)**-2H-1,2-benzothiazine 1,1-dioxide*** yellow powder, 51% yield, mp 133–135 °C. IR (KBr, cm^−1^): 1605 (CO), 1355, 1170 (SO_2_). ^1^H NMR (300 MHz, CDCl_3_) *δ* (ppm): 1.27 (brs, 2H, CH_2_C**H_2_**CH_2_), 1.98 (brs, 2H, C**H_2_**CH_2_CH_2_N-*piperazine*), 2.25 (brs, 4H of piperazine), 2.45 (s, 3H, CH_3_), 3.08–3.38 (m, 6H; 2H of CH_2_CH_2_C**H_2_** and 4H of piperazine), 7.01–8.21 (m, 12H, ArH), 15.70 (s, 1H, OH*_enolic_*). Anal. Calcd for C_30_H_30_F_3_N_3_O_4_S (585.64): C, 61.53; H, 5.16; N, 7.18; found: C, 61.83; H, 5.30; N, 7.06.

#### 2.2.2. Differential Scanning Calorimetry (DSC)

Calorimetric measurements were performed using a differential scanning calorimeter DSC 214 Polyma (Netzsch GmbH & Co., Selb, Germany) equipped with an Intracooler IC70 (Netzsch GmbH & Co., Selb, Germany) in the Laboratory of Elemental Analysis and Structural Research, Faculty of Pharmacy, Wroclaw Medical University.

For each sample, 2 mg of DPPC was dissolved in the appropriate amount of chloroform stock solution (5 mM) of the compounds studied (the compound/DPPC molar ratios in the samples were 0.06, 0.08, 0.10, and 0.12). All samples were dried under the stream of nitrogen and placed under a vacuum for at least 2 h to evaporate the chloroform (Rotary evaporator, Büchy Poland, Warsaw, Poland). In this process, the phospholipid was transferred onto the dry film on the inner surface of the Eppendorf tube. Samples were hydrated using 20 μL of Tris–EDTA buffer (pH 7.4). Hydrated mixtures of DPPC, the compounds studied, and buffer, closed in Eppendorf tubes, were heated (Labnet Dry Bath, Labnet International Inc., Edison, NJ, USA) to a temperature higher than 10 °C; the main phase transition temperature of the phospholipid was used (DPPC) and vortexed (neoVortex, neoLab, Ljubljana, Slovenia) until a homogeneous dispersion was obtained. Samples were sealed in aluminum pans type Concavus^®^ (Netzsch GmbH & Co., Selb, Germany). A pan of the same type, filled with 20 μL of Tris–EDTA buffer (pH 7.4), was employed as a reference. Measurements of the DPPC main phase transition were performed using the heat-flow measurement method at a heating rate of 1 °C per minute over a temperature range of 30–50 °C in a nitrogen dynamic atmosphere (25 mL/min). Data were analyzed off-line using Netzsch Proteus^®^ 7.1.0 (Netzsch GmbH & Co., Selb, Germany) analysis software. The transition enthalpies were stated in [J/g]. The measured heat was normalized per gram of lipid. The calorimeter was calibrated using standard samples from the calibration set 6.239.2–91.3.00 supplied by Netzsch (Netzsch GmbH & Co., Selb, Germany). All samples were weighed on a Sartorius CPA225D-0CE analytical balance (Sartorius AG, Gottingen, Germany) with a resolution of 0.01 mg.

#### 2.2.3. Fluorescence Spectroscopy

Unilamellar DPPC liposomes were obtained by sonification of 2 mM/L of phospholipid suspension in the same buffer solution as used in DSC experiments (pH 7.4) using a UP 200s sonificator (Dr. Hilscher, GmbH, Berlin, Germany).

Fluorescent dyes: laurdan and prodan stock solutions (1 mM) were prepared in DMSO. The stock solutions of the studied compounds (30 mM) were also prepared in DMSO. The dispersion of DPPC liposomes was incubated with the fluorescent dye in darkness for 30 min at room temperature, then the studied compound was added, and liposomes were incubated for another 20 min (also in darkness at room temperature). In all of the experiments, the final DPPC concentration was 200 μM. The concentration of the fluorescent dye (laurdan or prodan) was 5 μM. The studied compound concentration in the samples was 25–125 μM. The fluorescence experiments were performed with an LS 50B spectrofluorometer (Perkin-Elmer Ltd., Beaconsfield, UK) equipped with a xenon lamp using emission and excitation slits of 5 nm. The excitation wavelength for laurdan was 390 nm and for prodan was 360 nm. The recorded fluorescence spectra were processed with FLDM Perkin-Elmer 2000 software. It was investigated before the measurements that the studied compounds alone did not exhibit fluorescence in the spectral region of interest.

#### 2.2.4. Prediction of ADMET Properties

The physicochemical properties, pharmacokinetics, and ADMET activity of the designed meloxicam derivatives were estimated based on the comprehensive database ADMETlab (2.0) [23].

## 3. Results

### 3.1. Synthesis

The synthesis pathway of new meloxicam derivatives is presented in Figure 1. The key starting compound was commercially available saccharine **1**, which was condensed with 2-bromo-4′-methylacetophenone **2a** or 2-bromo-4′-methoxyacetophenone **2b** in a presence of triethylamine in dimethylformamide (DMF) at room temperature. The second step was the Gabriel–Colman rearrangement of compounds **3a** and **3b** through an opening of 1,2-thiazole ring in the presence of sodium ethanoate and closing with the formation of 1,2-thiazine ring, resulting in compounds **4a** and **4b**. The next step was the reaction of compound **4a** or **4b** with an appropriate piperazine derivative **5**, **6**, **7**, or **8** carried out in absolute ethanol with sodium ethanoate under reflux for 10 h. After the synthesis products were isolated from the mixture and crystallized from ethanol, the scheduled meloxicam derivatives **PR23**, **PR24**, **PR25**, **PR49**, and **PR50** were received with approximately 40–50% yields.

### 3.2. Differential Scanning Calorimetry (DSC)

The impact of the studied compounds on the lipid thermal behavior is presented in Figure 2, showing the example thermograms of DPPC mixed with **PR24** or **PR25** at different molar ratios. The two compounds differ only in the type of linker between the 1,2-benzothiazine scaffold and the phenylpiperazine moiety: compound **PR24** has a three-carbon chain and compound **PR25** has a two-carbon chain with a carbonyl group. Those thermograms show that the presence of an additional carbonyl group in **PR25** significantly affects and increases its interaction with the DPPC phospholipid bilayer in comparison to **PR24**.

The addition of the meloxicam analogues caused the vanishing of the DPPC pretransition and concentration-dependent shift of the main transition temperature towards lower values, accompanied by a decrease in the transition peaks area and the broadening of the peaks.

Moreover, at **PR25**/DPPC and different molar ratios (see Figure 2b), the main peak appears to be composed of two overlapping peaks with a shoulder. This proves that the main phase transition of the phospholipid is less cooperative when lipid is mixed with **PR25**.

The dependencies of the main transition temperature (T_M_), the transition peak width at half-height (ΔT_½_), and the transition enthalpy (ΔH) on the meloxicam analogue/lipid molar ratio obtained for the mixtures of DPPC with studied compounds are shown in Figure 3a–c, respectively.

All the examined compounds decreased the main transition temperature T_M_ of DPPC in a concentration-dependent manner (Figure 3a). The addition of studied compounds to DPPC also resulted in broadening of the transition peaks, which was visible as an increase in the transition width at half-height (Figure 3b). Moreover, all the examined compounds decreased the enthalpy (ΔH) of the DPPC main phase transition (Figure 3c). In case of all DPPC gel–liquid crystalline phase transition parameters, the most pronounced effects were found for compounds **PR25**, **PR49**, and **PR50**.

### 3.3. Fluorescence Spectroscopy

The addition of all studied compounds to DPPC liposomes incubated with a fluorescent probe resulted in quenching of laurdan or prodan fluorescence. The Stern–Volmer plots of the fluorescence quenching of both probes, and all the studied compounds, were linear (Figure 4).

The results of the quenching of fluorescence spectroscopy of two fluorescent probes (laurdan and prodan) may be grouped according to the similarities in the chemical structure of the tested meloxicam analogues. Compound **PR23** may be compared with **PR25** because they differ only in the substituent on the benzoyl group; **PR23** has a methoxy group and **PR25** has a methyl group. As can be seen in Figure 4b,d, the compound **PR25** has a more pronounced quenching effect on prodan, i.e., the methyl substituent seems to affect prodan fluorescence quenching more than the methoxy substituent. The **PR24** compound differs from the **PR25** in a carbon linker part and also has a weaker effect, so it may be presumed that a two-carbon linker with a carbonyl group in **PR25** enhances the effect of the compound. The **PR49** and **PR50** compounds also differ only in the type of linker between the 1,2-benzothiazine scaffold and the phenylpiperazine moiety: compound **PR50** has a three-carbon chain and compound **PR49** has a two-carbon chain with a carbonyl group. As can be seen in Figure 4e,f, the compound **PR49** has a more pronounced quenching effect on prodan fluorescence, which indicates that an additional carbonyl group is more preferred for interaction with the DPPC model membrane in gel phase (experiments were performed at room temperature) in the part of the phospholipid bilayer where prodan is located. Moreover, all the tested compounds quenched prodan fluorescence significantly more than meloxicam itself (see Figure 4a).

The quenching of prodan fluorescence was more pronounced than the quenching of laurdan fluorescence for all the tested compounds. According to Joseph Lakowicz, if the molecular location of the fluorescent probe within the lipid bilayer is known, quenching studies may be used to reveal the location of quenchers in the membrane [24]. The collisional quenching of fluorescence is described by Stern–Volmer plots: F_0_/F = 1 + K_D_ [Q]. where F_0_ and F are the fluorescence intensities in the absence and presence of the quencher, respectively, and Q is the concentration of the quencher. In dynamic quenching, K_D_ represents the Stern–Volmer constant. Linear Stern–Volmer plots indicate for one class of fluorophores which are equally accessible to the quencher. However, the observation of linear Stern–Volmer plots does not prove that fluorescence quenching is collisional. Two kinds of quenching—collisional and static—may be distinguished by measurements of the life-times of fluorescence. In collisional quenching, a concomitant decrease in the fluorescence intensity and lifetimes can be observed I_0_/I = τ_0_/τ. On the other hand, static quenching does not influence the fluorescence lifetimes [24]. However, in our studies, experiments on the measurements of fluorescence lifetimes were not carried out. Because the emission of fluorophore can be decreased both by collisional quenching and the formation of complexes, in such a case, an upward curvature of the plot towards the y axis is observed. In our experiments, such an upward curvature of the plots was observed in case of laurdan fluorescence, especially in the presence of meloxicam (Figure 4a) and in the case of prodan in the presence of **PR49** and **PR25** (Figure 4c). This means that in the case of these compounds, combined dynamic and static quenching took place.

It is known that prodan molecules locate closer to the hydrophilic surface of a bilayer [25] than laurdan, whose fluorophore is located closer to the phospholipid glycerol groups [26] (see Figure 5).

Therefore, we may conclude that all meloxicam analogues interacted with the DPPC phospholipid bilayer and that the bilayer region occupied by prodan was more affected by the presence of these compounds.

### 3.4. Prediction of ADMET Properties

It is very important that drug candidates exhibit strictly defined values of adsorption, distribution, metabolism, and excretion parameters. Additionally, the in vivo behavior of drug-like compounds can be influenced by simple physicochemical properties such as molecular weight (MW), the number of hydrogen bond donors (nHD) and acceptors (nHA), hydrophobicity, and polarity. Therefore, in the present paper, an ADMET analysis was performed to investigate the physicochemical properties and pharmacokinetic parameters of novel compounds based on the comprehensive database ADMETlab (2.0) [23]. The data obtained for the designed meloxicam analogues were compared with the properties of meloxicam [27,28].

The physicochemical properties of the compounds are presented in Table 1. As can be seen, the molecular weight of the compounds **PR23**, **PR24**, **PR25**, **PR49**, and **PR50** is much higher than the weight of meloxicam and ranges from 535 to 585 Da, which can result in worse values of absorption and elimination. However, it should be underlined that there is a criterion of weight (≥500) concerns to drugs that are not substrates for active transporters. All compounds have an optimal number of hydrogen bond acceptors (7–9) and donors (0). This is especially important in the case of oral drugs. Data showed that a high number of nHD influences the poor bioavailability and membrane permeability of small molecules [29]. According to the literature, the high values of the topological surface area (TPSA) can decrease membrane diffusion [30]. For example, the values of TPSA below 90 Å are required for compounds intended to cross the blood–brain barrier.

As presented in Table 1, the value of the TPSA of meloxicam derivatives does not exceed 104, which is favorable in terms of their permeability. It is known that high lipophilic agents can be trapped into bilayers. On the other hand, low lipophilicity can decrease the ability of drugs to the penetration of membranes [31]. According to these data, compounds **PR23**, **PR24**, **PR25**, **PR49**, and **PR50**, in comparison to meloxicam, are characterized by higher values of logP and logP at physiological pH. However, all compounds met Lipinski’s and Pfizer’s criteria (see Table 2).

What is important is that all compounds exhibit a good bioavailability and good or moderate blood–brain barrier permeability (BBB). On the other hand, they are characterized by a high plasma protein binding parameter (98–99%) indirectly related with the therapeutic index of drug-like compounds (see Appendix A).

The Caco-2 cell monolayer (colon adenocarcinoma cell lines) permeability model was used in the present paper to assess the absorption of the studied compounds. This model is experimentally used as an equivalent of the human intestinal epithelium to study the speed and permeability of the human digestive tract [32]. All designed compounds exhibit a good Caco-2 permeability and high passive MDCK (Madin−Darby canine kidney cells) permeability, which is the parameter describing the transporter-mediated mechanisms of penetration [33]. Data are presented in Appendix A.

Quite an important factor affecting the adsorption of drugs is their interactions with the membrane P-glycoprotein (P-gp). This protein is responsible for the transport of drugs out of the cells, which reduces their accumulation in the tissues [34]. This mechanism is largely responsible for the resistance of cancer cells to cytostatics (multidrug resistance), which is why the compounds that are P-gp inhibitors are sought [35]. The designed compounds are not a P-gp substrate and they exhibit a high probability of the inhibition of P-gp, which could be used in the future to overcome multidrug resistance in cancer therapy (see Appendix A).

It is obvious that metabolism plays a crucial role in the bioavailability of drugs. The most important class of phase I metabolism enzymes is cytochrome P450 enzymes (CYPs). The type of products of metabolic reactions often provide significant information about the efficacy of prodrugs, toxicities, and the clearance of drugs. All synthesized compounds are inhibitors of CYP2C9, which are the main enzyme-metabolized drugs with a narrow therapeutic index [36]. In contrast to meloxicam, all new compounds exhibit a high probability to inhibit CYP2C19, which is responsible for the activation of pro-carcinogens and the detoxification of some carcinogens [37]. Compounds **PR23**, **PR25**, and **PR49** and meloxicam were predicted to not be inhibitors of CYP2D6 (see Appendix A).

The predicted toxicity of the new compounds is at the level of the reference drug meloxicam (see Appendix A).

## 4. Discussion

It was previously shown in our laboratory that 1,2-benzothiazine (meloxicam) derivatives intercalate into the lipid bilayers and are located in the vicinity of the polar/apolar membrane interface (for EYPC liposomes) [38], or closer to the hydrophilic surface of a bilayer (for DPPC liposomes) [39]. The details of drug–lipid interactions depend, however, both on the lipid type as well as on the type of substitutions present at certain meloxicam derivative molecules.

As shown by the example thermograms presented in Figure 2, meloxicam analogues (**PR24** and **PR25**) eliminate the pretransition and substantially affect the main phase transition of DPPC in a concentration-dependent manner. Due to the fact that the pretransition is very sensitive to the presence of molecules that alter the packing of the phospholipid molecules and their hydration in gel state, its vanishing induced by meloxicam analogues is the first marker of the interaction of the studied compounds with the lipid bilayers. The addition of studied compounds evidently caused an alteration of the DPPC acyl chain packing in a gel state [40,41]. Moreover, the dependencies of the main transition temperature (T_M_), enthalpy change (∆H), and half-height width (∆T_½_) on the compound concentration, shown in Figure 3, allow us to conclude that the main phase transition of phospholipid was altered by meloxicam analogues more effectively than by meloxicam itself. According to the interpretation of calorimetric data proposed by Mahendra Kumar Jain and Nora Min Wu, the decrease in both the transition temperature and enthalpy change suggests that lipid polar heads as well as hydrocarbon chain regions were affected by the studied compounds. The character of the observed changes (decrease in T_M_ and ∆H and increase in ∆T_½_) allows us to also conclude that the interactions between lipid molecules in the gel state became weaker in the presence of studied compounds [42].

The thermal properties of meloxicam were previously investigated by Iwonne Kyrikou et al. and Sutapa Mondal Roy et al. It has been shown that meloxicam also lowered the T_M_ and widened the peak of the main phase transition, as well as abolished the pre-transition of the DPPC [43] or DMPC bilayer [44].

Spectrofluorimetric measurements confirmed that meloxicam analogues interact with the lipid bilayers. Fluorescence quenching by meloxicam, and other oxicams, has previously been demonstrated by Marlene Lúcio et al. [45]. In the present work, the quenching of the fluorescence of two probes, laurdan and prodan, by meloxicam analogues was investigated. It was found that the studied compounds quenched the fluorescence of prodan to a higher extent than laurdan in DPPC liposomes. It suggested that the regions close to the surface of the bilayer were more affected by the presence of the studied compounds than the regions near the phospholipid glycerol backbones. Additionally, the upward curvature of the Stern–Volmer plots of the quenching of prodan was noticed (see Figure 4). This suggested that the molecules of the studied compounds were present in the vicinity of the probe, quenching its fluorescence both by dynamic and static mechanisms. This effect was not observed for laurdan most likely due to the lesser accessibility of this probe for meloxicam analogues molecules that were present only in a small amount in the membrane region where laurdan fluorophore resided.

Analyzing the structure–activity relationship (SAR) of the tested compounds, it is clearly visible that the compounds containing the trifluoromethyl group (CF_3_) **PR49** and **PR50** have a more pronounced impact on the interaction with the model membranes (see Figure 3a,b). A very similar impact on the interaction with phospholipid membranes is also shown for the compound **PR25**, which does not contain the CF_3_ group, but instead a fluorine atom. However, the **PR25** compound has a two-carbon linker with a carbonyl group, and what is also known from our previous studies [38,39,46], compounds with this linker also interact more pronouncedly with phospholipid bilayers than compounds with a three-carbon linker without a carbonyl group (i.e., **PR24**).

In summary, all new meloxicam analogues interact more efficiently with model membranes than meloxicam alone. We may conclude that the introduction of modifications to the structure may have a positive effect on the penetration of new compounds through biological membranes.

Moreover, computational investigations of ADMET properties have shown that the new compounds are characterized by good physicochemical and drug-like properties. This thesis is mainly confirmed by the fulfillment of Lipinski’s and Pfizer’s criteria (see Table 2). However, we need to be aware that compounds meeting all the criteria are not necessarily orally bioavailable. Although the rule of five provide a reasonable degree of predictability of oral absorption, other models based on large chemical databases should be used. Therefore, absorption parameters were evaluated also based on Caco-2 and MDCK permeability models. As was presented, all the designed compounds exhibit a good Caco-2 permeability and a high passive MDCK permeability, which suggests the transporter-mediated mechanisms of penetration is an equally important parameter in the ability of the analyzed compounds to interact with P-glycoprotein. New meloxicam derivatives are not a Pgp substrate, and they exhibit a high probability of the inhibition of P-gp, which can be used in the future as a chemopreventive action, i.e., supporting cancer chemotherapy. To sum up, the obtained data allow us to indicate the safest and strongest drug candidates and exclude compounds that may fail in the next stages of drug development.

## 5. Conclusions

In the present work, we have synthesized five new meloxicam analogues and used calorimetry and fluorescence spectroscopy to study their interactions with DPPC phospholipid bilayers. We have shown that these interactions depend on the chemical structure of individual 1,2-bezothiazine derivatives. We may conclude that all the studied compounds alter the phospholipid bilayers properties. All the examined compounds decreased the main transition temperature T_M_ of DPPC and caused the broadening of the transition peaks in a concentration-dependent manner. Moreover, they also decreased the enthalpy (ΔH) of the DPPC main phase transition. The results of the experiments suggested that meloxicam analogues were most likely to interact with DPPC bilayer regions close to the membrane surface because they quenched the fluorescence of prodan to higher extents than for laurdan, which locates deeper in the phospholipid bilayer. We presume that the more pronounced intercalation of the studied compounds into the DPPC phospholipid bilayer may be related to the presence in the molecule of a two-carbon aliphatic linker with a carbonyl group and fluorine substituent/trifluoromethyl group (**PR25** and **PR49**), or the three-carbon linker together with a trifluoromethyl group (**PR50**).

## Data Availability

Not applicable.

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
