# Peer review of "New Meloxicam Derivatives—Synthesis and Interaction with Phospholipid Bilayers Measured by Differential Scanning Calorimetry and Fluorescence Spectroscopy"

_membranes, 2023, doi:10.3390/membranes13040416_

Round 1

Reviewer 1 Report

Comments are uploaded.

Author Response

Dear reviewer,

we are very grateful for your review of our manuscript. All changes in the manuscript are marked in yellow color. Please find our answers below:

1) The fluorescence measurements were used to examine the membrane partitioning of the studied compound. The static Stern-Volver quenching was used to estimate the probe location within the bilayer, as shown in Figs 4-5. These figures show clearly that the fluorescence quenching of Prodan reveals good linear dependence. However, the fluorescence quenching of Laurdan demonstrates non-linear behavior systematically, so there is some upward deviation from the static Stern-Volmer plot. This behavior is indicative of the mixed static-dynamics fluorescence quenching, which can also be analyzed by the modified Stern-Volver equation. The canonical book of Lakowicz (ref. 25) considers this case too. Therefore, it would be instructive to analyze in more detail the quenching results of Laurdan and the localization of the studied compounds.

When it comes to the quenching od fluorescence  of both markers (laurdan and prodan), looking at the graphs we do not see any differences in the linear course of the fluorescence intensity I0/I as a function of concentration, because the possible curvature of the graph for laurdan is within the standard deviation limit, which is depicted by vertical curves. If this is taken into account, then for prodan there is also a certain upward deviation of the graph visible for PR 25. In general, not only static (collision) but also dynamic extinction can occur here. To assess dynamic quenching, fluorescence lifetimes would have to be measured, which was not done in our laboratory. Both complex formation and dynamic quenching require the access of the quencher to the fluorophore, thus indirectly proves the location of the quencher in the membrane. We have added the following clarification to the discussion: „The collisional  quenching of fluorescence is described by Stern-Volmer plot F0/F =1+KD [Q] were F0 and F are fluorescence intensities in the absence and presence of the quencher, respectively and Q is concentration of quencher. In dynamic  quenching KD represents Stern-Volmer constant. A linear Stern-Volmer plots indicate for one class of fluorophores which are equally accessible to quencher. However, observation of linear Stern-Volmer plots does not proved that fluorescence quenching is collisional. Two kinds of quenching -  collisional and static  may be distinguished by measurements of life-times  of fluorescence. In collisional quenching it is observed a concomitant decrease of fluorescence intensity and lifetimes I0/I =τ0/τ. On the other hand static quenching does not influence fluorescence lifetimes [25]. However, in our studies experiments on measurements of fluorescence lifetimes were not carried out. Because emission of fluorophore can be decreased both by collisional quenching and formation of complexes in such a case an upward curvature of the plot towards axis y is observed.  In our experiments such upward curvature of the plots was observed in case of laurdan fluorescence especially in the presence of meloxicam (Fig. 4a) and in case of prodan in the presence PR49 and PR25 (Fig. 4c). It means that in case of these compounds combined dynamic and static quenching took place.”

2) It would also be instructive to provide raw fluorescence spectra for Figs. 4-5 in Supplementary Materials.

We fully agree with the reviewer that it is good practice to include fluorescence spectra in supplementary materials. Unfortunately, the version of the spectroscopic software that we use for measurements is old and it does not co-work with other types of computer software and does not allow us to export files (only ongoing processing). So we would have to use print screens which resolution is unsatisfactory, and that is the reason why we did not do it.

3) Finally, the manuscript title is somewhat misleading. The title promises some computational investigations. In fact, the authors utilized one-click ADMET properties calculations using the standard tools. These findings are not computational investigations. For the studied systems, some real computational investigations would be using molecular dynamics simulations to model the probe binding and partitioning into a bilayer at atomistic resolutions.

As suggested by the reviewer, we have changed the title to: „New meloxicam derivatives – synthesis and interaction with phospholipid bilayers measured by differential scanning calorimetry and fluorescence spectroscopy.”

Reviewer 2 Report

The manuscript is devoted to the study of the meloxicam analogues to interact with phospholipid bilayers. I believe that this manuscript is interesting and should be publishable in this journal; however there are several scientific aspects of this manuscript that I feel the authors must first address.

1. The title should be changed; Pleas reconsider title to be more clear-it is too general.

2. line 44-47 This section does not provide complete information about the membrane models. Please, addition more information and references.

3. line 66-68 “2 Phosphatidylcholine is abundant in the mucous membrane of the digestive tract, where it has a protective effect on the mucosa” should be added references this aspect.

4. Please, clarify the choice of the lipids composition DPPC, but not DOPC, POPC or DMPC.

5. The measurement temperature of different methods should be indicated. Especially, in the paragraph 2.2.3 due to the DPPC temperature transition about 41C.

6. What is the purity of the samples meloxicam derivatives?

7. Please, clarify the choice of the concentrations of meloxicam derivatives. In this regard, it is necessary to discuss the detergent effects of compound on the properties of lipid membranes.

8. Figure 2 presented only two derivatives. Please, clarify the absence of DPPC termogram in the presence of other meloxicam derivatives and most importantly, how does meloxim itself affect the DPPC melting tremogram.

9. Figure 3 presented the main transition temperature (TM), the transition peak width at half-height (ΔT½) and the transition enthalpy (ΔH). At the same time, on the figure indicated the T½ and Figure 3b and 3c dimensions not specified.

10. Please, clarify the deconvolution of the DPPC peak in the presence of the derivative PR25.

11. Please, restructured the paragraph 3.3 – it is more convenient to compare figure 4 and figure 5 on the same panel and, probably, addition the summarizing this data in the table.

12. What did the authors want to show in Figure 6 – literature data of the localization of fluorescent probes or localization of the sensitive probes and meloxicam derivatives? Then the figure does not indicate the localization of derivative molecules. Therefore Figure 6 therefore should be modified.

13. Please, clarify the toxicity of meloxicam derivatives.

14. It is known that meloxicam induced the membrane fusion. What the action mechanism authors suggest for these meloxicam derivatives - the inhibition of the membrane fusion also? This aspect is not discussed. Why?

15. The Moreover, my great concern is related to the conclusions. They are too short and schematic. They should be modified. The most important findings of this work should be supported by results and their biological significance should be clearly specified.

Author Response

Dear reviewer,

we are very grateful for your review of our manuscript. All changes in the manuscript are marked in yellow color. Please find our answers below:

  1. The title should be changed; Please reconsider title to be more clear-it is too general.

As suggested by the reviewer, we have changed the title to: „New meloxicam derivatives – synthesis and interaction with phospholipid bilayers measured by differential scanning calorimetry and fluorescence spectroscopy.”

  1. line 44-47 This section does not provide complete information about the membrane models. Please, add more information and references.

As suggested by the reviewer, we have revised this part of the manuscript

  1. line 66-68 “2Phosphatidylcholine is abundant in the mucous membrane of the digestive tract, where it has a protective effect on the mucosa” should be added references this aspect.

The reference has been added.

  1. Please, clarify the choice of the lipids composition DPPC, but not DOPC, POPC or DMPC.

We performed our study using DPPC bilayers due to the fact that we used this phospholipid before.  Thanks to this, the results of this work and several previous, in which we studied the impact of other 1,2-benzothiazine derivatives, can be easily compared. This allows us to determine the possible impact of differences in the structure of the compounds we synthesize on interactions with model membranes.

  1. The measurement temperature of different methods should be indicated. Especially, in the paragraph 2.2.3 due to the DPPC temperature transition about 41C.

Perhaps the temperature scale on the thermogram is not very clear, but the temperature read by the calorimeter was in accordance with the literature (for DPPC 41,5 ⁰C). The calorimeter was calibrated using standard samples from calibration set 6.239.2-91.3.00 supplied by Netzsch (Netzsch GmbH & Co.,Selb, Germany).

  1. What is the purity of the samples meloxicam derivatives?

To determine the purity of the newly synthesized compounds, we used the spectroscopic method - proton nuclear magnetic resonance spectroscopy (1H NMR) and elemental analysis (C, H, N). The 1H NMR spectra show all protons present in the tested sample (spectra of all tested compounds are included in the Supplementary data). As you can see in the spectra, the compounds are completely pure, because apart from the protons of the tested compound, there are no other peaks in the spectrum indicating the presence of impurities. The 1H NMR technique confirms both the purity of the compounds and their structure, which confirms the planned meloxicam derivatives.

The second method - elemental analysis shows whether the percentage composition of elements in the tested compound corresponds to their content calculated theoretically. For example, for the PR23 compound, the calculated carbon content is 60.97%, while the content obtained after testing is 60.75%, similarly for hydrogen atoms (calculated H, 4.75%; found 4.68%) and nitrogen atoms (calculated N, 7.62%; found 7.42%). A compound is considered pure when the content of the element obtained in the test does not differ by more than +/- 0.4% from the theoretical (calculated) value. All new compounds meet this requirement, which proves their analytical purity. All data for new compounds are provided in section 2.2. Experimental - 2.2.1. Synthesis and in Supplementary data.

  1. Please, clarify the choice of the concentrations of meloxicam derivatives. In this regard, it is necessary to discuss the detergent effects of compound on the properties of lipid membranes.

When determining the concentrations of the tested compounds (molar ratios of the test compound: phospholipid), we were guided by the concentrations in which drugs belonging to the oxicam group were previously tested, because we use them as reference compounds (it was most often the molar ratio of 0.1), as well as our previous studies, so that the results could be compare (these were molar ratios of 0.06, 0.08, 0.1, 0.12).

  1. Figure 2 presented only two derivatives. Please, clarify the absence of DPPC thermogram in the presence of other meloxicam derivatives and most importantly, how does meloxicam itself affect the DPPC melting thremogram.

The thermograms are shown as examples. We wanted to show that a slight difference in the chemical structure of these compounds (the presence of one carbonyl group in PR25, and lack of it in PR24) caused a significant difference in the impact on thermotropic properties. We did not show the thermograms of the control (meloxicam) because these results have been previously published in the literature: DOI: 10.1016/j.chemphyslip.2004.06.005.

  1. Figure 3 presented the main transition temperature (TM), the transition peak width at half-height (ΔT½) and the transition enthalpy (ΔH). At the same time, on the figure indicated the T½ and Figure 3b and 3c dimensions not specified.

The figure 3 presents the influence of studied compounds on the parameters of DPPC main phase transition. Fig. 3a is  the main transition temperature, Fig. 3b is peak width at half-height  and 3 c is enthalpy change. We hope that it is visible.

  1. Please, clarify the deconvolution of the DPPC peak in the presence of the derivative PR25.

We believe that the carbonyl group contained in PR25 may affect the cooperativity of DPPC bilayer and possibly causes partial phase separation.

  1. Please, restructured the paragraph 3.3 – it is more convenient to compare figure 4 and figure 5 on the same panel and, probably, addition the summarizing this data in the table.

Figures 4 and 5 have been changed as requested by the reviewer. The results of fluorescence quenching of laurdan and prodan by meloxicam, which is the reference compound, are on the same page as the results of the test compounds.

  1. What did the authors want to show in Figure 6 – literature data of the localization of fluorescent probes or localization of the sensitive probes and meloxicam derivatives? Then the figure does not indicate the localization of derivative molecules. Therefore Figure 6 therefore should be modified.

The caption under the drawing was expanded so that it was clear where the tested compounds were located.

  1. Please, clarify the toxicity of meloxicam derivatives.

The toxicity of the new meloxicam derivatives tested in this work (PR23, PR24, PR25, PR49 and PR50) has not yet been determined. However, toxicity studies were performed with very similar derivatives in rat stomachs and were found to be completely non-toxic (0 points on a scale of 0-5 points) compared to piroxicam (3 points). This was published in an article entitled: Synthesis and pharmacological evaluation of novel arylpiperazine oxicams derivatives as potent analgesics without ulcerogenicity (DOI: 10.1016/j.bmc.2019.03.007). For other similar derivatives, in vitro toxicity studies have been performed on normal human dermal fibroblasts - NHDF (DOI: 10.1016/j.bioorg.2020.104476). On this basis, we presume that the new meloxicam derivatives will also prove to be non-toxic. Conducting these studies is planned in the long term.

  1. It is known that meloxicam induced the membrane fusion. What the action mechanism authors suggest for these meloxicam derivatives - the inhibition of the membrane fusion also? This aspect is not discussed. Why?

In our team, we synthesize new compounds, often derivatives of existing drugs, and test their properties, often checking whether they differ from the reference drug. Our main goal, however, is to understand how the modification of the chemical structure of new compounds affects their properties. It was not our purpose to test membrane fusion and we did not investigate this issue.

  1. The Moreover, my great concern is related to the conclusions. They are too short and schematic. They should be modified. The most important findings of this work should be supported by results and their biological significance should be clearly specified.

We have changed the conclusions part in the manuscript.

Round 2

Reviewer 1 Report

The authors addressed my concerns properly. Further review is not needed.  I can recommend the manuscript for publication in its present form.

Reviewer 2 Report

The authors considered all my comments. The text has been significantly improved and the manuscript can be recommended for publication in its current form.